# Antibody levels and protection after Hepatitis B vaccine in adult vaccinated healthcare workers in northern Uganda

**Moses Ocan**[1]*, **Frances Acheng**[2], **Carol Otike**[3], **Judith Beinomugisha**[4], **David Katete**[5], **Celestino Obua**[6]

**1** Department of Pharmacology & Therapeutics, Makerere University College of Health Sciences, Kampala, Uganda, **2** Infectious Disease Institute, College of Health Sciences, Yumbe General Hospital, Makerere University, Yumbe District, Kampala, Uganda, **3** Clinical Epidemiology Unit, Makerere University College of Health Sciences, Kampala, Uganda, **4** Makerere University Lung Institute, Kampala, Uganda, **5** Department of Immunology and Molecular Biology, Makerere University College of Health Sciences, Kampala, Uganda, **6** Department of Pharmacology and Vice Chancellor, Mbarara University of Science and Technology, Mbarara, Uganda

* ocanmoses@gmail.com

**Data Availability Statement:** All relevant data are within the paper. Requests for additional information should be addressed to the

## Abstract

Hepatitis B vaccine has contributed to the reduction in hepatitis B virus infections and chronic disease globally. Screening to establish extent of vaccine induced immune response and provision of booster dose are limited in most low-and-middle income countries (LMICs). Our study investigated the extent of protective immune response and breakthrough hepatitis B virus infections among adult vaccinated healthcare workers in selected health facilities in northern Uganda. A cross-sectional study was conducted among 300 randomly selected adult hepatitis B vaccinated healthcare workers in Lira and Gulu regional referral hospitals in northern Uganda. Blood samples were collected and qualitative analysis of Hepatitis B surface antigen (HBsAg), Hepatitis B surface antigen antibody (HBsAb), Hepatitis B envelop antigen (HBeAg), Hepatitis B envelop antibody (HBeAb) and Hepatitis B core antibody (HBcAb) conducted using ELISA method. Quantitative assessment of anti-hepatitis B antibody (anti-HBs) levels was done using COBAS immunoassay analyzer. Multiple logistic regression was done to establish factors associated with protective anti-HBs levels ($\geq 10$ mIU/mL) among adult vaccinate healthcare workers at 95% level of significance. A high proportion, 81.3% (244/300) of the study participants completed all three hepatitis B vaccine dose schedules. Two (0.7%, 2/300) of the study participants had active hepatitis B virus infection. Of the 300 study participants, 2.3% (7/300) had positive HBsAg; 88.7% (266/300) had detectable HBsAb; 2.3% (7/300) had positive HBeAg; 4% (12/300) had positive HBeAb and 17.7% (53/300) had positive HBcAb. Majority, 83% (249/300) had a protective hepatitis B antibody levels ($\geq 10$ mIU/mL). Hepatitis B vaccine provides protective immunity against hepatitis B virus infection regardless of whether one gets a booster dose or not. Protective immune response persisted for over ten years following hepatitis B vaccination among the healthcare workers.

corresponding author and data may be provided on reasonable request.

**Funding:** Research reported in this publication was supported by the Fogarty International Center of the National Institutes of Health, U.S. Department of State's Office of the U.S Global AIDS Coordinator and Health Diplomacy (S/GAC), and President's Emergency Plan for AIDS (PEPFAR) under Award Number IR25TW011213. The content is solely the responsibility of the authors and does not necessarily represent the official views of the National Institutes of Health.

**Competing interests:** The authors have declared that no competing interests exist.

## Introduction

Elimination of hepatitis B virus (HBV) transmission is an achievable public health goal, particularly in the light of proven effectiveness and safety of hepatitis B vaccine [1]. Studies conducted in areas with high HBV endemicity have demonstrated declines in the prevalence of chronic HBV among children to < 2% after routine infant vaccination [1]. A substantial decline in HBV-related disease burden and prevalence of chronic HBV infection has been observed among children following introduction of universal infant hepatitis B vaccination [2]. However, the vaccine may not provide protection from exposure to hepatitis B virus later on in life due to waning of immune memory over time [3].

Persistence of hepatitis B antibodies (anti-HBs) and ability of the immune system to mount a response to exposure of HBV later in life is necessary for long term protection against hepatitis B virus infection [4]. Some studies have confirmed persistence of antibodies and immune memory following hepatitis B vaccination [5] while others confirm waning of antibody concentrations 13–15 years after primary vaccination among those vaccinated at birth [6]. The current vaccination of adult individuals against hepatitis B virus is premised on the fact that sufficient anti-HBs concentrations and immune memory is formed against HBV. However, unless routine post-vaccination serological testing is performed, it remains unclear what proportion of individuals who complete all 3 dose schedules of hepatitis B vaccine actually get protected, as was found in an earlier study where 22.9% of vaccinated children had undetected antibody levels [7].

Annually 5.9% of the healthcare workers are exposed to HBV corresponding to 66,000 preventable HBV infections globally [8]. Healthcare workers thus represent an important group in the population that need to be protected against HBV infection. While hepatitis B is a disease of public health concern in Uganda [9], healthcare workers (HCWs) in the country are at higher risk for transmission of hepatitis B virus (HBV) compared to the general population [10]. They have a higher risk of contracting the disease from exposure (eye, oral mucosa and skin) to potentially infectious patient's blood and percutaneously from contaminated sharp objects such as needles [11].

Northern Uganda has disproportionately high prevalence of hepatitis B virus infection compared to other parts of the country [12]. As a preventive measure, all healthcare workers in Uganda are required to take hepatitis B vaccine as adults. A recent study by Ssekamatte *et al.*, [13] reported a rather low hepatitis B vaccine dose completion of 57.8% only among healthcare workers in central Uganda. In spite of the low hepatitis B vaccine dose completion rate, no evaluation of hepatitis B vaccination has been done to assess immune response to the vaccine since the introduction of hepatitis B vaccination in Uganda in 2002. This could be due to the limited funding of the health sector, a common occurrence in most LMICs [14]. There is paucity of information on the extent of immune protection against hepatitis B virus infection among adult vaccinated individuals in Uganda despite the high risk of exposure. We therefore set out to determine the proportion of healthcare workers with protective levels of anti-HBs and also evaluate the prevalence of hepatitis B virus infection among those without protective anti-HBs post hepatitis B vaccination.

## Methods

### Ethical considerations

Ethical review and approval of the protocol was done by the Makerere University School of Biomedical Science Research and Ethic Review Committee (#SBS-REC 798). Additionally, administrative clearance was obtained from the hospitals prior to study initiation. A written

informed consent was obtained from potential study participants prior to enrollment into the study.

## Study design, setting and population

This was a cross sectional study done in Lira and Gulu regional referral hospitals in northern Uganda from October to December 2020. In Uganda, regional referral hospitals offer specialist clinical services such as psychiatry, Ear, Nose and Throat (ENT), ophthalmology, higher level surgical and medical services, and ancillary services (laboratory, medical imaging and pathology). They also provide general healthcare services including, preventive, promotive, curative, maternity, in-patient health services, surgery, blood transfusion, laboratory and medical imaging services. The regional referral hospitals are also involved in teaching and research. Hepatitis B screening and management services are also offered in regional referral hospitals in Uganda. Gulu and Lira regional referral hospital has 320, 420 healthcare workers respectively.

The study was conducted among adult vaccinated healthcare workers in Lira and Gulu regional referral hospitals in northern Uganda. We enrolled healthcare workers who had previously received primary Hepatitis B vaccine irrespective of when the vaccination was received.

## Sample size determination

The sample size was calculated using Kish Leslie formula [15] applying a prevalence of non-immune protection of 22.9% [7], 95% level of significance and 10% non-response giving a sample size of 300 study participants.

## Data collection

Interview data collection: The interview was conducted by two research assistants, a laboratory technologist and a nurse. The two research assistants were trained on the survey tool prior to conducting the interviews. Using simple random sampling, the healthcare workers in Lira and Gulu regional referral hospitals were approached for inclusion into the study. A sample frame of healthcare workers in each of the hospital was obtained from the hospital administrator. The name of each healthcare worker was then written on a separate piece of small paper which was folded and placed in a basket. With shaking of a basket at each point, one piece of paper was picked at a time without replacement until the required sample size was obtained. A healthcare worker whose name was picked was then approached for recruitment into the study. A written informed consent was obtained prior to enrollment into the study. All the healthcare workers who reported having taken a hepatitis B vaccine and consented for the study were recruited. Healthcare workers who were under hepatitis B treatment were excluded. Interview data was collected using interviewer administered questionnaires, which had been pretested on 10 healthcare workers at Mulago national referral hospital. The study tool collected data on, (i) socio-demographic characteristics, (ii) risk of exposure to HBV, (iii) HBV testing, (iv) HBV vaccine awareness, and (v) HBV vaccination (S1 Appendix).

Laboratory data collection: For each consenting healthcare worker, 4mls of venous blood was collected using ethylene diamine tetra acetic acid (EDTA) vacutainer tubes. The blood samples were immediately centrifuged (3000rpm) for five minutes and plasma separated from blood cells in to cryovials. The plasma was then screened using HBV Combo Rapid test (Vaxpert Inc Suite 355 Two South Biscaynne Blvd. Miami, FI, USA). This is a rapid test for qualitative detection of; Hepatitis B surface antigen (HBsAg) a protein on the surface of hepatitis B virus whose presence in serum is an indicator of acute or chronic hepatitis B virus infection; Hepatitis B surface antibody (HBsAb or Anti-HBs) a protein produced by the body's immune

system in response to the presence of Hepatitis B surface antigen; Hepatitis B envelop antigen (HBeAg) a viral protein made by the hepatitis B virus that is released from infected liver cells into the blood and is an indicator of active HBV replication; Hepatitis B envelop antibody (HBeAb) a protein produced by the body's immune system in response to HBeAg a marker of resolution of illness; and Hepatitis B core antibody (HBcAb or Anti-HBc) a protein produced by the body's immune system in response to hepatitis B virus and is an indicator of previous hepatitis B virus infection. e. The test was done following the manufacture's guidelines. Briefly, the test cassette was removed from the sealed foil pouch and placed on a clean, leveled surface work-top in the laboratory. Holding the dropper vertically, three (3) full drops (approximately 75μl) of plasma was transferred to each sample well and the timer started. The results were read after 15 minutes. The appearance of a colored line in the control region (C) confirmed the viability of the test. For HBsAg, HBsAb and HBeAg tests, a positive result was confirmed with the presence of two distinct colored lines on the test cassette, one being the test region and the other the control region. While for HBeAb and HBcAb tests, a positive result was confirmed by presence of one colored line in the control region (C) and no colored line in the test region (T). Sample tests with no colored line in the control region (C) were repeated. In addition, known positive and negative control samples were run alongside the test samples for quality control. The plasma samples with detectable HBsAb were then transferred to Uganda Blood Bank Nakasero laboratory under 20˚C, for analysis of HBsAb concentration. The COBAS Elecsys 2010 immunoassay analyzer was used in the analysis of HBsAb concentration. The analyzer was calibrated using control and sample plasma prior to the analysis. 50μl of plasma for each sample was processed in duplicate following manufacturer's guidelines, the reaction mixture is then aspirated into the measuring cell. The HBsAb concentration was measured by comparing the electro-chem-luminescence signal obtained to that from the calibration. Hepatitis B antibody levels $\geq$ 10mIU/mL were considered protective [16].

### Data management and analysis

At the end of each data collection day all the questionnaires were checked for completeness. Double data entry was done by two data entrants (OC and KR) into Epi-Data *ver 3.1*. Data was transferred to STATA *ver* 23 and cleaned prior to analysis. Categorical variables were analyzed using a modified Poisson regression with robust standard errors [17, 18]. Bivariate analysis was performed for each of the independent variables to determine whether they were independently associated with immune response to hepatitis B vaccine using prevalence ratios (PR) and *p*-values at 95% level of significance. All variables were entered and carried to the multivariate logistic regression using a backward elimination method. Confounding was assessed by comparing crude and adjusted PR, with a difference between crude and adjusted PR of greater than 10% considered as confounding.

## Results

### Socio-demographic characteristics of study participants

A total of 300 participants were sampled from Gulu and Lira regional referral hospital. Most participants were females (52%), aged between 26–35 years (43.3%), and were either nurses or midwives (45.5%). The main form of hepatitis B infection risk was IM/IV injection (86%). Most of the study participants (96.7%) received their last dose of hepatitis B vaccine shot between 2010 and 2020. Majority of the participants 244/300 (81.3%) completed all the three doses of hepatitis B vaccine. A high proportion (96.7%) of the study participants received Hepatitis B vaccine at 0, 1, 6 vaccine schedules (months). Majority (99%) did not receive a booster doze of hepatitis B vaccine (Table 1). Of the 300 study participants, 2.3% (7/300) had positive

**Table 1. Descriptive characteristics of healthcare workers (n = 300) in Gulu and Lira regional referral hospitals, October–December 2020.**

| Characteristic | Description | Frequency, N = 300 n (%) |
|---|---|---|
| Age (years) | 18–25 | 89 (29.7) |
| | 26–35 | 130 (43.3) |
| | 36+ | 81 (27.0) |
| Sex | Male | 144 (48.0) |
| | Female | 156 (52.0) |
| Specialty | Doctor/Clinician/physician | 17 (5.7) |
| | Nurse/Midwife | 136 (45.5) |
| | Pharmacist | 3 (1.0) |
| | Laboratory technician | 70 (23.4) |
| | Others | 73 (24.5) |
| Risk exposure | Needle prick injury | 58 (19.3) |
| | Blood splash | 77 (25.7) |
| | Blood transfusion | 10 (3.3) |
| | Injection (IM/IV) | 258 (86.0) |
| | Conducted surgery | 82 (27.3) |
| | Had a tattoo on the skin | 3 (1.0) |
| | Had a surgical operation | 33 (11.0) |
| | Others | 2 (0.7) |
| Year hepatitis B vaccine intake | 2000–2009 | 10 (3.33) |
| | 2010–2020 | 290 (96.7) |
| Age at last hepatitis B vaccination | 18–25 | 161 (53.7) |
| | 26–35 | 102 (34.0) |
| | 36+ | 37 (12.3) |
| Number of doses received | 1 | 20 (6.7) |
| | 2 | 36 (12.0) |
| | 3 | 244 (81.3) |
| Completed hepatitis vaccine dose | Yes | 244 (81.3) |
| | No | 56 (18.7) |
| Time interval (months) | 0, 1, 12 | 1 (0.4) |
| | 0, 1, 6 | 236 (96.7) |
| | 0, 2, 4 | 2 (0.8) |
| | 0, 3, 6 | 4 (1.6) |
| | Do not recall | 1 (0.4) |
| Received a booster dose | Yes | 7 (2.3) |
| | No | 293 (97.7) |

HBsAg; 88.7% (266/300) had positive HBsAb; 2.3% (7/300) had positive HBeAg; 4% (12/300) had positive HBeAb and 17.7% (53/300) had positive HBcAb.

## Hepatitis B vaccine immune response among health workers

Of the 300 hepatitis B vaccinated healthcare workers screened, 270 (90%) had detectable anti-HBs, while 30 (10%) did not have detectable hepatitis B antibodies (HBsAb), two of whom were found to have active hepatitis B infection. Of the 270 who had detectable anti-HBs, antibody titre test was performed and the test was successful in 266 blood samples, where 93.6% (249/266, 95%CI: 89.9–96) had protective antibody titre (> 10mIU/mL) against hepatitis B virus infection.

Of the 266 participants with detectable anti-hepatitis B antibodies, 6.4% (17/266, 95%CI: 4–9.6) did not have protective (<10mIU/mL) antibody titre (non-responders), 15.4% (41/266, 95%CI: 11.5–20.3) had 10-99mIU/mL antibody titre (poor responders), 49.6% (132/266, 95% CI: 43.6–55.6) had 100-999mIU/mL antibody titre (moderate responder) and 28.6% (76/266, 95%CI: 23.4–34.3) had ≥1000mIU/mL antibody titre (strong responders). Overall, we found that the majority, 72% (216/300) of study participants had protective hepatitis B immunity (≥10mIU/mL) due to hepatitis B vaccine (HBsAg$^{-ve}$, HBcAb$^{-ve}$, HBsAb$^{+ve}$). Furthermore, one in six vaccinated healthcare workers (51/300, 17%) had protective hepatitis B immunity (≥10mIU/mL) due to a natural infection (HBsAg$^{-ve}$, HBcAb$^{+ve}$, HBsAb$^{+ve}$). A third of the study participants are susceptible to hepatitis B virus infection (HBsAg, and HBcAb, HBsAb all negative). Some of the participants (2/300, 0.7%) had active hepatitis B virus infection (HbsAg$^{+ve}$, HbcAb$^{+ve}$, HbsAb$^{-ve}$).

## Factors associated with hepatitis B virus immune response among healthcare workers in Gulu and Lira regional referral hospitals

In bivariate analyses, the factors that were significantly associated with protective anti-HBs levels include, year of last hepatitis B vaccine dose ($p<0.001$), time schedule for hepatitis B vaccine (0-1-6 months) ($p<0.001$), and booster hepatitis B vaccine dose ($p<0.001$) (Table 2).

Majority of the participants, 50% (133/266) who were 15–25 years at the time of vaccination had sufficient immune response (≥10mIU/mL).

**Table 2. Factors associated with hepatitis B immune response among vaccinated healthcare workers (n = 266) in Gulu and Lira regional referral hospitals, October-December 2021.**

| Characteristic | Description | Immune response (≥10mIU/mL) n (%) | cPR | 95% CI | aPR | 95%CI | P-value |
|---|---|---|---|---|---|---|---|
| **Age (years)** | 18–25 | 74 (27.8) | 1 | | 1 | | |
| | 26–35 | 110 (41.4) | 1.00 | 0.93–1.08 | 0.99 | 0.92–1.07 | 0.854 |
| | 36+ | 65 (24.4) | 0.99 | 0.91–1.08 | 0.93 | 0.79–1.09 | 0.357 |
| **Sex** | Female | 133 (50) | 1 | | 1 | | |
| | Male | 116 (43.6) | 1.01 | 0.95–1.08 | 1.06 | 0.98–1.14 | 0.162 |
| **Specialty** | Allied Health | 25 (9.4) | 1 | | 1 | | |
| | Doctors | 14 (5.3) | 0.91 | 0.74–1.11 | 0.86 | 0.66–1.13 | 0.277 |
| | Nurse/Midwife | 116 (43.6) | 1.01 | 0.85–1.08 | 1.04 | 0.94–1.15 | 0.463 |
| | Pharmacist | 2 (0.8) | 0.69 | 0.31–1.55 | 0.51 | 0.13–2.03 | 0.341 |
| | Lab. Technician | 56 (21.1) | 0.94 | 0.84–1.05 | 0.96 | 0.85–1.08 | 0.500 |
| | Others | 35 (13.2) | 0.96 | 0.85–1.08 | 0.99 | 0.88–1.13 | 0.992 |
| **Year vaccine intake** | 2000–2009 | 9 (3.4) | 1 | | 1 | | |
| | 2010–2020 | 240 (90.2) | 0.93 | 0.90–0.96 | 0.94 | 0.9–0.97 | **<0.001** |
| **Age at vaccination** | 15–25 | 133 (50) | 1 | | 1 | | |
| | 26–35 | 85 (31.9) | 0.97 | 0.90–1.04 | 1.02 | 0.92–1.14 | 0.693 |
| | 36+ | 31 (11.7) | 1.03 | 0.95–1.11 | 1.10 | 0.93–1.31 | 0.264 |
| **Vaccine schedule** (months) | 0, 1,12 | 1 (0.4) | 1 | | 1 | | |
| | 0, 1, 6 | 202 (75.9) | 0.94 | 0.91–0.97 | 1.07 | 1.0–1.11 | **<0.001** |
| | 0, 2, 4 | 2 (0.8) | 1 | 1 | | | 1.000 |
| | 0, 3, 6 | 1 (0.4) | 1 | 1 | | | 1.000 |
| | Do not recall | 1 (0.4) | 1 | 1 | | | 1.000 |
| **Received booster** | No | 246 (92.5) | 1 | | | | |
| | Yes | 3 (1.1) | 1.07 | 1.04–1.10 | 1.07 | 1.0–1.11 | **<0.001** |

After adjusting for other covariates in multivariable analysis, there was no predictor of participants having protective anti-HBs titre (>10mIU/mL). There was no significant difference in the odds of having protective anti-HBs titre (>10mIU/mL) in participants who received the vaccine between 2000–2009 and those who received the vaccine between 2010–2020 (aPR = 0.94, p<0.001). The odds of having protective anti-HBs titre (>10mIU/mL) did not significantly differ by vaccine schedule (aPR = 1.07, p<0.001) and receiving of booster dose (aPR = 1.07, p<0.001) (Table 2).

## Discussion

In this study we found a fairly high hepatitis B vaccine three-dose completion rate (81.3%) among healthcare workers in northern Uganda. This was an improvement when compared to a previous report of 57.8% among healthcare workers in central Uganda and another report from other LMICs with completion rates of 40–90% (16–19) [13]. The high Hepatitis B vaccine completion rate found in this study could be due to the higher prevalence of Hepatitis B virus infection in northern Uganda compared to the rest of the country [12]. Whereas there is an increase in public awareness through promotion of hepatitis B vaccine and the requirement by the ministry of health for all healthcare workers to receive hepatitis B vaccine prior to engagement in clinical work including all health professional students [9], our findings indicate that the vaccination rate still falls short of the 100% completion among healthcare workers as recommended by the WHO [19]. The likelihood of spread of hepatitis B virus infection between healthcare workers and their patients become more apparent especially given the unknown hepatitis B vaccine coverage in the general population in Uganda. Of concern was the increased risk of transmission where we found that the majority of the healthcare workers got intramuscular/intravenous injections, or conducted surgical procedures and accidentally had blood splashed on their bodies during those procedures, further emphasizing the need for completion of hepatitis B vaccination doses.

On qualitative analysis, we found a high proportion of healthcare workers with detectable anti-HBs indicative of either recovery from hepatitis B infection or immunity secondary to hepatitis B vaccination. After quantification of the anti-HBs, we found over 90% of the healthcare workers with detectable anti-HBs had protective antibody concentration ($\geq$ 10mIU/mL). These findings are similar to reports from previous studies [20] that reported presence of protective immune response among individuals vaccinated with hepatitis B vaccine. The high proportion of healthcare workers with detectable anti-HBs is a confirmation of Hepatitis B vaccination among the study participants. In addition, protective anti-HBs titers found in majority (72%) of the study participants is an indicator of effective immune response to the hepatitis B vaccine.

The findings that healthcare workers who received hepatitis B vaccine 20 years back as adults still had protective anti-HBs levels is an indicator of long-term protection against hepatitis B virus infection even though the majority never received a booster vaccine dose. While the Uganda Ministry of Health does not conduct periodic screening of healthcare workers to assess for immune protection, our findings in this study are in line with evidence from previous studies [20, 21] that reported more 30 years of immune protection from hepatitis B vaccine among individuals vaccinated as children and young adults. Our findings show that regular screening to assess extent of hepatitis B immune protection among individuals who have completed all the three hepatitis B vaccine dose schedules may not be necessary due to the limited resources in most LMICs. However, for the few individuals who had weak or no protective immune response (<10mIU/mL) against HBV following complete vaccination, provision of a challenge dose of the vaccine is recommended as this has previously been shown to prompt development of immunity[22].

While only three of the healthcare workers had received a booster dose of the hepatitis B vaccine, we found no significant difference in the presence of protective immune response between individuals who received a booster dose and those who did not. A similar finding was reported in a previous study by Bruce *et al*. [20] which found development of protective immune response ($\geq$10mIU/mL) among fully vaccinated individuals who did not receive a booster hepatitis B vaccine dose. In our study we included individuals who reported to have received hepatitis B vaccines in the past 20 years. Although there is currently no screening to establish immune response to hepatitis B vaccine in Uganda, it is likely that despite the booster dose of the vaccine, vaccinated individuals already had sufficiently high levels of anti-HBs in the body. This could be due to the persistence of anti-hepatitis B antibodies in vaccinated individuals [22]. Our findings indicate that completion of all three hepatitis B vaccine doses is sufficient for the development of protective anti-HBs titers.

A low proportion (2.3%) of healthcare workers in this study had detectable HBsAg an indicator of acute or chronic hepatitis B virus infection, while 1 in every 6 healthcare workers had a positive HBcAb indicative of previous or ongoing hepatitis B virus infection, possibly due to non-response to the hepatitis B vaccine as all study participants had been vaccinated. This is supported by the findings that a low proportion of hepatitis B virus infection occurred among healthcare workers who had undetectable anti-HBs on qualitative screening. The lack of protective anti-HBs titre following vaccination found in our study is similar to findings from a previous study [7].

While our study had some limitations where the HBV Combo Rapid test was used for qualitative screening of HBsAg, HBsAb, HBeAg, HBeAb and HBcAb, we were able to transfer samples with positive HBsAb to the national blood bank laboratory (Nakasero blood bank) for quantitative measurement of the HBsAb levels in the plasma samples. Additionally, we also referred participants with positive HBsAg for further tests to confirm Hepatitis B viral infection. It was also not possible to ascertain from the study participants what kind of hepatitis B vaccine that they had received, whether plasma derived or recombinant vaccine. However, results from previous studies have shown that antibody response and the effectiveness of the plasma-derived vaccine are similar to that of recombinant hepatitis B vaccine [19, 23].

## Conclusion and recommendation

Most vaccinated healthcare workers in northern Uganda developed protective anti-HBs levels and the protective immune response persisted for over ten years following vaccination. Some vaccinated healthcare had weak/no immunity and developed breakthrough hepatitis B virus infection which would require booster hepatitis B vaccination. There may be need for post-vaccination testing to assess immunological priming from hepatitis B vaccine among healthcare workers.

## Supporting information

**S1 Appendix. Study data collection tool.**
(PDF)

## Acknowledgments

We would like to acknowledge the research participants and the hospital administration of Lira and Gulu regional referral hospitals for participating in the study and allowing us to conduct the study in the hospitals respectively.

## Author Contributions

**Conceptualization:** Moses Ocan, Judith Beinomugisha, David Katete.

**Formal analysis:** Moses Ocan, Carol Otike, Celestino Obua.

**Methodology:** Moses Ocan, Frances Acheng.

**Writing – original draft:** Moses Ocan.

**Writing – review & editing:** Moses Ocan, Frances Acheng, Carol Otike, Judith Beinomugisha, Celestino Obua.

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
