## [Decision Letter · Decision Letter 0]

6 Sep 2021

PONE-D-21-17608Antibody levels and prevalence of hepatitis B infection among vaccinated healthcare workers in health facilities in northern UgandaPLOS ONE

Dear Dr. Ocan,

Thank you for submitting your manuscript to PLOS ONE. After careful consideration, we feel that it has merit but does not fully meet PLOS ONE’s publication criteria as it currently stands. Therefore, we invite you to submit a

We look forward to receiving your revised manuscript.

Kind regards,

Ray Borrow, Ph.D., FRCPath

Academic Editor

PLOS ONE

2. Thank you for stating in the text of your manuscript "A written informed consent was obtained prior to enrollment into the study. ". Please also add this information to your ethics statement in the online submission form.

3. Please include additional information regarding the survey or questionnaire used in the study and ensure that you have provided sufficient details that others could replicate the analyses. For instance, if you developed the survey or questionnaire as part of this study and it is not under a copyright more restrictive than CC-BY, please include a copy, in both the original language and English, as Supporting Information. If the questionnaire is published, please provide a citation to the (1) questionnaire and/or (2) original publication associated with the questionnaire.

5. Please amend the manuscript submission data (via Edit Submission) to include author David Katete.

Additional Editor Comments (if provided):

Reviewers' comments:

Reviewer's Responses to Questions

**Comments to the Author**

1. Is the manuscript technically sound, and do the data support the conclusions?

Reviewer #1: Yes

Reviewer #2: Yes

Reviewer #3: Yes

2. Has the statistical analysis been performed appropriately and rigorously? 

Reviewer #1: Yes

Reviewer #2: No

Reviewer #3: Yes

3. Have the authors made all data underlying the findings in their manuscript fully available?

Reviewer #1: Yes

Reviewer #2: Yes

Reviewer #3: Yes

4. Is the manuscript presented in an intelligible fashion and written in standard English?

Reviewer #1: Yes

Reviewer #2: No

Reviewer #3: Yes

5. Review Comments to the Author

Reviewer #1: The paper is well written and is very important in the field of Hep B prevention and control.

I would combine tables 3 and 4 together so one can compare both crude and adjusted effects easitly.

I was also keen to see the explanation of what the recent vaccnations 2010-2020 are least likely to have protective anti-HB titres. One would think that recent vaccination might be associated with protection as opposed those vaccinated a long time ago.

Could it be the type of vaccine used is diffrent, or what was picked up as high titres for older vaccination would be a boost from natural infection.

Reviewer #2: Firstly, I would like to congratulate the authors for addressing such an important public health topic with a relevant methodology. Generally, the paper is well written and structured. However, further efforts are required to make the study more attractive to the readers. I have provided and raised numerous comments and questions on the manuscript that has to be addressed by the authors. Authors can get my comments and questions from my attachment.

Reviewer #3: General comments: I would like to thank the authors for this wonderful article. It addresses a public health concern that is to some extent ignored in many low resource settings. The article also builds on previous work undertaken in Uganda. The paper is well-written. Nonetheless, there are a few issues that the authors need to address to make the paper more scientifically informative.

1) Methods: Provide sufficient detail on the study setting. Where are the study facilities located? How are the staffing levels, cadres and services provided? Do the healthcare facilities provide Hepatitis B screening and management services?

2) Methods: Why was Mulago national referral hospital chosen for the pre-test? Is Mulago hospital and the two study healthcare facilities comparable?

3) A section on quality assurance and control for both serological testing and interviews with health workers is missing. Were the research assistants trained?

4) Discussion: Generally, the authors need to strengthen the discussion. Whereas they have attempted to provide implications for the findings, there is no explanation for most of the findings. For example, what explains the high proportion of healthcare workers with detectable anti-HBs?

5) Discussion: The authors point out that there was an improvement in the hepatitis B vaccination rate compared to a previous study by Ssekamatte et al. Both studies are recent and were conducted in Uganda. What explains the improvement? Are there any contextual differences between healthcare facilities in Wakiso and Lira and Gulu hospitals? What explains the high vaccine completion rates in Gulu and Lira referral hospitals? Remember that regional referral hospitals also have treatment centres for Hepatitis B.

6) Discussion: The authors mentioned that ‘’ While only three of the healthcare workers had received a booster dose of the hepatitis B vaccine, we found no significant difference in the presence of protective immune response between individuals who received a booster dose and those who did not’’. However, the authors do not provide an explanation to the findings. Why is it so?

7) Details of the ethics approval are lacking. Kindly provide information on the ethics clearance for the current study.

6. PLOS authors have the option to publish the peer review history of their article (what does this mean?). If published, this will include your full peer review and any attached files.

Reviewer #1: **Yes: **Michelo Simuyandi

Reviewer #2: **Yes: **Tefera Alemu

Reviewer #3: No

---

## [Author Response · Author response to Decision Letter 0]

28 Sep 2021

RESPONSE TO REVIEWERS COMMENTS ON MANUSCRIPT PONE-D-21-17608

Editor’s comment: Thank you for stating in the text of your manuscript "A written informed consent was obtained prior to enrollment into the study. ". Please also add this information to your ethics statement in the online submission form.

Response: We are grateful for the comment, the statement "A written informed consent was obtained prior to enrollment into the study” has been added to the ethics statement in the online submission form. 

Ethical considerations

Ethical review and approval of the protocol was done by the Makerere University School of Biomedical Science Research and Ethic Review Committee (#SBS-REC 798). Additionally, administrative clearance was obtained from the hospitals prior to study initiation. A written informed consent was obtained from potential study participants prior to enrollment into the study. 

Editor’s comment: Please include additional information regarding the survey or questionnaire used in the study and ensure that you have provided sufficient details that others could replicate the analyses. For instance, if you developed the survey or questionnaire as part of this study and it is not under a copyright more restrictive than CC-BY, please include a copy, in both the original language and English, as Supporting Information. If the questionnaire is published, please provide a citation to the (1) questionnaire and/or (2) original publication associated with the questionnaire.

Response: Thanks for the comment, the questionnaire used for data collection in the study has been added as supplementary material in the revised manuscript (Additional file_1). 

Editor’s comment: We note that the grant information you provided in the ‘Funding Information’ and ‘Financial Disclosure’ sections do not match.

Response: The details of the grant has been provided. However, I am not the primary recipient of the grant which provided financial support for this study.

Editor’s comment: Please amend the manuscript submission data (via Edit Submission) to include author David Katete.

Response: Thanks for the observation, David Katete has been added as an author in the manuscript submission system. 

Comments Reviewer #1: The paper is well written and is very important in the field of Hep B prevention and control.

I would combine tables 3 and 4 together so one can compare both crude and adjusted effects easitly.

Response: Thanks for the comment, tables 3 and 4 have been combined to enable easy comparison of crude and adjusted odds ratios in the revised manuscript 

Comments Reviewer #1: I was also keen to see the explanation of what the recent vaccnations 2010-2020 are least likely to have protective anti-HB titres. One would think that recent vaccination might be associated with protection as opposed those vaccinated a long time ago.

Could it be the type of vaccine used is diffrent, or what was picked up as high titres for older vaccination would be a boost from natural infection.

Response: We agree with the concern raised by the reviewer, the adjusted odds ratio of 0.94 is close to 1.0 and therefore the interpretation of this result was incorrect. We have adjusted the interpretation in the revised manuscript, this is now interpreted as no significant difference in the odds of immune protection between individuals who received the hepatitis vaccine between 2000-2009 and 2010-2020. 

Comments Reviewer #2: Firstly, I would like to congratulate the authors for addressing such an important public health topic with a relevant methodology. Generally, the paper is well written and structured. However, further efforts are required to make the study more attractive to the readers. I have provided and raised numerous comments and questions on the manuscript that has to be addressed by the authors. Authors can get my comments and questions from my attachment.

Response: Thanks for the compliment, the comments have been addressed in the revised manuscript 

Comments Reviewer #2: In my opinion, it is better if you modify your title to make it easily understandable, attention grasping and in line with your study objectives. 

Response: Thanks for the comment, the title of the manuscript has been adjusted in the revised manuscript. The current title is “Antibody levels and protection after Hepatitis B vaccine in adult vaccinated healthcare workers in northern Uganda”. This has been incorporated in the revised manuscript. 

Comments Reviewer #2: Your sample size should be mentioned in the abstract section in between line number 28 and 29.

Response: Thanks for the comment, the sample size of 300 healthcare workers has been mentioned in the abstract of the revised manuscript. 

Comments Reviewer #2: You put too many information under abstract section. Results that answer your objectives are enough.

Response: Thanks, some of the information has been removed from the abstract of the revised manuscript. 

Comments Reviewer #2, Introduction section: You stated that “However, the vaccine may not provide protection from exposure to hepatitis B virus later on in life due to waning of immune memory over time (3).” This idea needs a strong scientific evidence and it should be cited with additional and credible source to convince the readers.

Response: Additional references have been provided in the revised manuscript

1. Cocchio S, Baldo V, Volpin A, Fonzo M, Floreani A, Furlan P, et al. Persistence of Anti-Hbs after up to 30 Years in Health Care Workers Vaccinated against Hepatitis B Virus. Vaccines. 2021; 9:323.

2. Bruce MG, Bruden D, Hurlburt D, Zanis C, Thompson G, Rea L, et al. Antibody Levels and Protection After Hepatitis B Vaccine: Results of a 30-Year Follow-up Study and Response to a Booster Dose. The Journal of Infectious Diseases. 2016; 214:16-22.

Comments Reviewer #2, Methods and Materials: The tittle in line number 105 should be “Method and Materials”, instead of method.

Response: Thanks, the title “Methods and Materials” has been incorporated in the revised manuscript

Comments Reviewer #2, Methods and Materials: Your sampling technique is not stated

Response: Simple random sampling was used and is stated under the section on data collection. A sample frame of healthcare workers in the hospital was obtained from the hospital administrator. The name of each healthcare worker was then written on a separate piece of small paper which was folded and placed in a basket. With shaking of a basket at each point, one piece of paper was picked at a time without replacement until the required sample size was obtained. 

Comments Reviewer #2, Methods and Materials: No exclusion or inclusion criteria?

Response: Thanks for the comment, we included all healthcare workers who reported having taken a hepatitis B vaccine. Healthcare workers who were under hepatitis B treatment were not included in the study. 

Comments Reviewer #2, Methods and Materials: The following terms should be defined under method section. Otherwise, it will be difficult to understand the results.

Hepatitis B surface antigen (HBsAg), Hepatitis B surface antibody (HBsAb), Hepatitis B envelop antigen (HBeAg), Hepatitis B envelop antibody (HBeAb), and Hepatitis B core antibody (HBcAb).”

Response:

Hepatitis B surface antigen (HBsAg)- a protein on the surface of hepatitis B virus; it can be detected in high levels in serum during acute or chronic hepatitis B virus infection. The presence of HBsAg indicates that the person is infectious

Hepatitis B surface antibody (HBsAb or Anti-HBs)- a protein produced by the body’s immune system in response to the presence of the surface protein of hepatitis B virus (HBsAg)

Hepatitis B envelop antigen (HBeAg)- This is a viral protein made by the hepatitis B virus that is released from infected liver cells into the blood and is an indicator of active HBV replication. 

Hepatitis B envelop antibody (HBeAb)- a protein produced by the body’s immune system in response to HBeAg. The development of HBeAb during acute Hepatitis B infection is a marker of resolution of illness

Hepatitis B core antibody (HBcAb or Anti-HBc)- a protein produced by the body’s immune system in response to hepatitis B virus. This antibody does not provide any protection or immunity against the hepatitis B virus. A positive or "reactive" test indicates that a person may have been infected with the hepatitis B virus at some point in time.

Comments Reviewer #2, Methods and Materials: Line number 148 “Hepatitis B antibody levels ≥ 10mIU/mL were considered protective.” The recommendation for this cut off value should be cited.

Response: A reference for the protective Hepatitis B antibody levels has been provided in the revised manuscript. 

Jack AD, Hall AJ, Maine N, Mendy M, Whittle HC (1999): What Level of Hepatitis B Antibody Is Protective? The Journal of Infectious Diseases; 179:489–92

Poovorawan Y, Chongsrisawat V, Theamboonlers A, Leroux-Roels G, Kuriyakose S, Leyssen M, et al. Evidence of protection against clinical and chronic hepatitis B infection 20 years after infant vaccination in a high endemicity region. J Viral Hepat. 2011; 18:369-75.

Comments Reviewer #2, Results:

Which one is your research objective? The first or the second sentence? Is immune development status due to vaccination and natural infection the same? 

“Where 93.6% (249/266, 95%CI: 89.9-96) had protective immune response (> 10mIU/mL) against hepatitis B virus.” 

“Overall, we found that the majority, 72% (216/300) of the study participants had protective hepatitis B immunity 192 (≥10mIU/mL) due to hepatitis B vaccine.”

Response: The extent of development of protective immune response from natural infection is not the same as that from vaccination. In our study, 93.6% (249/266) of the study participants developed protective immune response (> 10mIU/mL) from both natural infection and vaccination. Of the individuals with protective immune response (> 10mIU/mL), 72% (216/300) was due to hepatitis B vaccine. Therefore, the second statement “Overall, we found that the majority, 72% (216/300) of the study participants had protective hepatitis B immunity 192 (≥10mIU/mL) due to hepatitis B vaccine” is the main objective of our study. This has been clarified in the revised manuscript. 

Comments Reviewer #2, Results: In line number 192and 193 you said that “due to hepatitis B vaccine” and “due to a natural infection”. These should be defined under definition of terms of the methodology part and readers can understood easily it meaning. 

Response: ‘due to natural infection’- The Hepatitis B antibodies produced by the body as a result of hepatitis B virus infection.

‘due to hepatitis B vaccine’- The Hepatitis B antibodies produced by the body as a result of Hepatitis B vaccine. 

This has been incorporated in the revised manuscript 

Comments Reviewer #2, Results: You have duplication of ideas in between line number 184 up to 187.

Response: This has been adjusted in the revised manuscript

Comments Reviewer #2, Results: All the points in table 2 are described under the paragraphs above. So, what is the importance of this table? I think it is duplications of efforts and the interpretations in table 2 are ideas to be mentioned under definition of terms/operational definition. You can simply remove the last column (frequency) and put it under method section as operational definition or you can modify it.

Response: Thanks for the comment, we agree the information in table 2 has been incorporated and provided in a descriptive form in the revised manuscript. 

Comments Reviewer #2, Results: In line number 206 and 207, you stated “Factors associated with hepatitis B virus immune response among 207 healthcare workers in Gulu and Lira Regional Referral hospitals”. It is too long and seems table tittle. 

Results: Thanks for the comment, this is a section header and has been adjusted in the revised manuscript 

Comments Reviewer #2, Results: There is no need to prepare different tables for bivariate and multivariate regressions models. Please merge table 3 and 4.

Response: Thanks for the comment, table 3 and 4 have been merged into one in the revised manuscript

Comments Reviewer #2, Results: In multivariate analysis, your two-by-two table did not fulfill the Chi-square assumption for regression. There are empty cell and your CI also includes 1, but your p value is less than 5%. How can it be possible? I don’t think so that you have a significant predictor.

Response: In our analysis, we used modified poisson regression which does not set to full fil the chi-square assumption as required in chi-square test. Using poisson regression helps take care of the small and empty cells. In addition to addressing convergence when all the observations are small or empty cells, poisson regression is also useful when the prevalence of the outcome of interest is high. 

1. HUANG, F. L. 2019. Alternatives to logistic regression models in experimental studies. The Journal of Experimental Education, 1-16.

2. ZOU, G. 2004. A modified poisson regression approach to prospective studies with binary data. Am J Epidemiol, 159, 702-6.

Comments Reviewer #2, Results: Your statement under table 4 is difficult to understand, for instance “were 7% times” and the likes. The paragraph from line number 119 up to 229 are also not easily understandable.

Response: This has been adjusted in the revised manuscript 

Comments Reviewer #2, Results: Be consistent throughout. Example, for your dependent variable, use either “protective anti-HBs levels” or “hepatitis B immune response” throughout.

Response: Thanks for the comment, we have adhered to using ‘protective anti-HBs levels’ throughout the revised manuscript

Comments Reviewer #2, Recommendation: In your recommendation, you did not talk about the factors of protective anti-HBs levels. Is that not your study objective from the outset or not?

Response: This has been adjusted in the revised manuscript

Comments Reviewer #2: Grammar and punctuation errors are her and there in the paper and sometimes your idea is difficult to understand and not coherent. Please get assistance from someone who is fluent in English language and/or manuscript editing software’s.

Results: Thanks for the comment, this has been adjusted in the revised manuscript 

Comments Reviewer #3: General comments: I would like to thank the authors for this wonderful article. It addresses a public health concern that is to some extent ignored in many low resource settings. The article also builds on previous work undertaken in Uganda. The paper is well-written. Nonetheless, there are a few issues that the authors need to address to make the paper more scientifically informative.

Response: Thanks for the compliment 

Comments Reviewer #3, Methods: Provide sufficient detail on the study setting. Where are the study facilities located? How are the staffing levels, cadres and services provided? Do the healthcare facilities provide Hepatitis B screening and management services?

Response: The study was conducted in Lira and Gulu regional referral hospitals. In Uganda, regional referral hospitals offer specialist clinical services such as psychiatry, Ear, Nose and Throat (ENT), ophthalmology, higher level surgical and medical services, and ancillary services (laboratory, medical imaging and pathology). They also provide general healthcare services including, preventive, promotive, curative, maternity, in-patient health services, surgery, blood transfusion, laboratory and medical imaging services. The regional referral hospitals are also involved in teaching and research. Hepatitis B screening and management services are also offered in regional referral hospitals in Uganda. 

Government of Uganda. Health sector strategic and investment plan- Promoting people’s health to enhance socio-economic development 2010/11-2014/15. Kampala: Ministry of Health; 2010.

Comments Reviewer #3, Methods: Why was Mulago national referral hospital chosen for the pre-test? Is Mulago hospital and the two study healthcare facilities comparable?

Response: Mulago hospital is a national referral hospital and is not comparable to the regional referral hospitals in terms of the extent and level of services offered. However, both Mulago national referral hospital and the two study healthcare facilities provide Hepatitis B screening, treatment and vaccination services.

Comments Reviewer #3, Methods: A section on quality assurance and control for both serological testing and interviews with health workers is missing. Were the research assistants trained?

Response: The interview was conducted by two research assistants, a laboratory technologist and a nurse. The two research assistants were trained on the survey tool prior to conducting the interviews. 

The description of quality control measures for both the interviews and laboratory experiments have been incorporated in the revised manuscript. 

Comments Reviewer #3, Discussion: Generally, the authors need to strengthen the discussion. Whereas they have attempted to provide implications for the findings, there is no explanation for most of the findings. For example, what explains the high proportion of healthcare workers with detectable anti-HBs?

Response: The high proportion of healthcare workers with detectable anti-HBs is a confirmation of Hepatitis B vaccination among the study participants. In addition, protective anti-HBs titers were found in majority (72%) of the study participants an indicator of effective immune response to the hepatitis B vaccine. 

This has been incorporated in the discussion section of the revised manuscript. 

 Comments Reviewer #3, Discussion: The authors point out that there was an improvement in the hepatitis B vaccination rate compared to a previous study by Ssekamatte et al. Both studies are recent and were conducted in Uganda. What explains the improvement? Are there any contextual differences between healthcare facilities in Wakiso and Lira and Gulu hospitals? What explains the high vaccine completion rates in Gulu and Lira referral hospitals? Remember that regional referral hospitals also have treatment centres for Hepatitis B.

Response: Thanks, we agree with the comment. Contextual factors may explain the higher vaccine completion rate in Lira and Gulu compared to Wakiso. For example, northern Uganda has the highest prevalence of hepatitis B virus infection in the country and this is likely to have driven more healthcare workers in the region to take up the hepatitis B vaccine compared to the rest of the country. 

This has been incorporated in the revised manuscript 

Comments Reviewer #3, Discussion: The authors mentioned that ‘’ While only three of the healthcare workers had received a booster dose of the hepatitis B vaccine, we found no significant difference in the presence of protective immune response between individuals who received a booster dose and those who did not’’. However, the authors do not provide an explanation to the findings. Why is it so?

Response: A recent study by Cocchio et al., 2021 showed that Hepatitis B antibodies persist in vaccinated healthcare workers for up to 30 years. In our study we included individuals who reported to had received hepatitis B vaccines in the past 20 years (2000 – 2020) it is thus likely that despite the booster dose of the vaccine, vaccinated individuals still had sufficiently high levels of anti-HBs in the body.

This has been incorporated in the revised manuscript. 

Comments Reviewer #3: Details of the ethics approval are lacking. Kindly provide information on the ethics clearance for the current study.

Response: Ethical review and approval of the protocol was done by the Makerere University School of Biomedical Science Research and Ethic Review Committee (#SBS-REC 798). Additionally, administrative clearance was obtained from the hospitals prior to study initiation. A written informed consent was obtained from potential study participants prior to enrollment into the study. 

This has been incorporated in the adjusted manuscript.

---

## [Decision Letter · Decision Letter 1]

17 Dec 2021

Antibody levels and protection after Hepatitis B vaccine in adult vaccinated healthcare workers in northern Uganda

PONE-D-21-17608R1

Dear Dr. Ocan,

We’re pleased to inform you that your manuscript has been judged scientifically suitable for publication and will be formally accepted for publication once it meets all outstanding technical requirements.

Kind regards,

Ray Borrow, Ph.D., FRCPath

Academic Editor

PLOS ONE

Additional Editor Comments (optional):

Reviewers' comments:

Reviewer's Responses to Questions

**Comments to the Author**

1. If the authors have adequately addressed your comments raised in a previous round of review and you feel that this manuscript is now acceptable for publication, you may indicate that here to bypass the “Comments to the Author” section, enter your conflict of interest statement in the “Confidential to Editor” section, and submit your "Accept" recommendation.

Reviewer #1: All comments have been addressed

2. Is the manuscript technically sound, and do the data support the conclusions?

Reviewer #1: Yes

3. Has the statistical analysis been performed appropriately and rigorously? 

Reviewer #1: Yes

4. Have the authors made all data underlying the findings in their manuscript fully available?

Reviewer #1: Yes

5. Is the manuscript presented in an intelligible fashion and written in standard English?

Reviewer #1: Yes

6. Review Comments to the Author

Reviewer #1: (No Response)

7. PLOS authors have the option to publish the peer review history of their article (what does this mean?). If published, this will include your full peer review and any attached files.

Reviewer #1: **Yes: **Michelo Simuyandi

---

## [Editor Report · Acceptance letter]

23 Dec 2021

PONE-D-21-17608R1 

Antibody levels and protection after Hepatitis B vaccine in adult vaccinated healthcare workers in northern Uganda 

Dear Dr. Ocan:

I'm pleased to inform you that your manuscript has been deemed suitable for publication in PLOS ONE. Congratulations! Your manuscript is now with our production department. 

Kind regards, 

on behalf of

Prof. Ray Borrow 

Academic Editor

PLOS ONE